# Pro-Inflammatory Interactions of Dolutegravir with Human Neutrophils in an In Vitro Study

**DOI:** 10.3390/molecules27249057

**Published:** 2022-12-19

**Authors:** Annette J. Theron, Ronald Anderson, Morris Madzime, Theresa M. Rossouw, Helen C. Steel, Pieter W. A. Meyer, Moloko C. Cholo, Luyanda L. I. Kwofie, Charles Feldman, Gregory R. Tintinger

**Affiliations:** 1Department of Immunology, Faculty of Health Sciences, University of Pretoria, Pretoria 0002, South Africa; 2Department of Immunology, Tshwane Academic Division, National Health Laboratory Services, Pretoria 0002, South Africa; 3Department of Internal Medicine, Faculty of Health Sciences, University of the Witwatersrand, Johannesburg 2193, South Africa; 4Department of Internal Medicine, Steve Biko Academic Hospital, Faculty of Health Sciences, University of Pretoria, Pretoria 0002, South Africa

**Keywords:** cytosolic calcium, elastase, HIV, ionophore, pro-inflammatory activity

## Abstract

There is increasing awareness of an association between the uptake of the HIV integrase inhibitor, dolutegravir, in first-line antiretroviral regimens with unusual weight gain and development of the metabolic syndrome, particularly in African women. Although seemingly unexplored, the development of systemic inflammation linked to the putative pro-inflammatory activity of dolutegravir represents a plausible pathophysiological mechanism of this unusual weight gain. This possibility was explored in the current study undertaken to investigate the effects of dolutegravir (2.5–20 μg/mL) on several pro-inflammatory activities of neutrophils isolated from the blood of healthy, adult humans. These activities included the generation of reactive oxygen species (ROS), degranulation (elastase release) and alterations in the concentrations of cytosolic Ca^2+^ using chemiluminescence, spectrophotometric and fluorimetric procedures, respectively. Exposure of neutrophils to dolutegravir alone resulted in the abrupt, dose-related, and significant (*p* < 0.0039–*p* < 0.0022) generation of ROS that was attenuated by the inclusion of the Ca^2+^-chelating agent, EGTA, or inhibitors of NADPH oxidase (diphenyleneiodonium chloride, DPI), phospholipase C (U733122), myeloperoxidase (sodium azide) and phosphoinositol-3-kinase (wortmannin). In addition, exposure to dolutegravir augmented the release of elastase by stimulus-activated neutrophils. These pro-inflammatory effects of dolutegravir on neutrophils were associated with significant, rapid, and sustained increases in the concentrations of cytosolic Ca^2+^ that appeared to originate from the extracellular compartment, seemingly consistent with an ionophore-like property of dolutegravir. These findings are preliminary and necessitate verification in the clinical setting of HIV infection. Nevertheless, given the complex link between inflammation and obesity, these pro-inflammatory interactions of dolutegravir with neutrophils may contribute to unexplained weight gain, possibly via the development of insulin resistance.

## 1. Introduction

Integrase strand transfer inhibitors (INSTIs) represent a relatively new class of antiretroviral agents that act by inhibiting the incorporation of HIV-1 pro-viral DNA into the host genome, a step that is necessary for viral replication [1]. Dolutegravir is a second-generation INSTI that displays a superior barrier to resistance, is highly effective, is generally well tolerated, and has been recommended by the WHO as the preferred first- and second-line treatment for all populations [2,3]. However, a side effect affecting the class of INSTIs, in general, and dolutegravir in particular, is excessive weight gain with or without the development of the metabolic syndrome during the course of treatment; this side effect has been described in some studies [4,5,6,7,8], but not others. Although the pathophysiology underpinning this potentially harmful weight gain remains unexplained, mechanisms linked to adipose tissue dysfunction and fibrosis, immune dysfunction and inflammation, as well as dysbiosis and microbial translocation, have been proposed [9]. Obesity results in the pathological expansion of adipose tissue that drives excessive lipid accumulation in adipocytes, which elicits the infiltration of immune/inflammatory cells, possibly secondary to hypoxia [10]. In this context, the possible involvement of dolutegravir as a trigger of inflammation-associated obesity merits investigation.

Neutrophils, the most abundant type of circulating leukocyte, play an essential role in the body’s innate immune response to infection [11]. Neutrophils control infectious pathogens by phagocytosing and degrading microbes using an antimicrobial arsenal that includes a variety of toxic reactive oxygen species (ROS) and proteolytic enzymes [12]. However, as these products are indiscriminate by nature, they have the capacity to inflict bystander tissue damage in the setting of an over-exuberant neutrophil response [13]. In this context, it is important to note that obesity has been associated with the activation of innate immune responses, specifically neutrophil activation [14,15,16].

The effects of various antiretroviral drugs on neutrophil function have recently been reviewed by Madzime et al. [17]; however, no data are yet available for dolutegravir. In the context of other cell types, prolonged exposure to dolutegravir in vitro has been shown to cause hemolysis of erythrocytes, while a derivative of this agent has been found to be cytotoxic for non-small cell lung cancer cells [18,19]. In both cases, cell death is associated with an influx of extracellular Ca^2+^ and intracellular oxidative stress [18,19]. The current study investigated the effects of exposure of isolated human neutrophils to dolutegravir on the pro-oxidative/pro-inflammatory activities of these cells, as well as a possible association with alterations in cellular cytosolic Ca^2+^ concentrations. Based on our observations, a putative link between HIV infection, dolutegravir therapy, neutrophil activation, and obesity is proposed.

## 2. Results

### 2.1. ROS Production

The effects of dolutegravir (2.5–10 μg/mL) on the spontaneous generation of ROS using the lucigenin- and luminol-enhanced chemiluminescence (CL) procedures are shown in Figure 1. Treatment of the cells with dolutegravir resulted in a dose-related stimulation of the generation of ROS, which was particularly evident in the case of luminol-enhanced CL (*p* = 0.0022). Dolutegravir (5–10 µg/mL) also caused significant spontaneous activation of superoxide production as measured by lucigenin-enhanced CL (*p* = 0.0039). The maximum stimulation was observed at 10 μg/mL of dolutegravir, resulting in a 2.52-fold and a 60-fold median increase in the generation of ROS by neutrophils with the lucigenin-enhanced and luminol-enhanced chemiluminescence procedures, respectively.

The effects of 10 μg/mL of dolutegravir on luminol-enhanced CL were almost completely abolished by inclusion of EGTA in the cell-suspending medium or pretreatment of neutrophils with DPI, sodium azide, U73122 or wortmannin. The respective median percentages inhibition of dolutegravir-mediated oxidant generation in the presence of EGTA or the various pharmacological inhibitors were 98.2 (98:99.2), 99.5 (99.4:99.7), 96.6 (95.4:97.1), 99.9 (99.96:99.98) and 99.96 (99.93:99.97), (*n* = 2–4 donors with 6–11 replicates). The attenuation of the CL responses by DPI and azide demonstrates that dolutegravir is a true activator of the neutrophil NADPH oxidase and myeloperoxidase release, respectively, seemingly by a Ca^2+^-dependent mechanism. This contention is supported by the inhibitory effects of EGTA and U73122 on these dolutegravir-activated neutrophil activities, while those of wortmannin also suggest the involvement of phosphoinositide 3-kinase.

In an additional series of control experiments, the addition of dolutegravir (10 µg/mL) to HBSS + luminol in the absence of neutrophils did not alter CL, the values for the systems without and with dolutegravir being 19 (9:35) and 12 (8:23) RLU, respectively (data from five experiments).

### 2.2. Elastase Release

The effects of dolutegravir on the release of elastase from unstimulated neutrophils, as well as from cells activated with FMLP/cytochalasin B (CB), are shown in Figure 2. Treatment of the cells with dolutegravir resulted in statistically significant, dose-related stimulation of the release of elastase from unstimulated cells and FMLP/CB-stimulated cells (*p* < 0.0000), albeit of greater magnitude in the case of the latter. In both cases, the maximum stimulation was seen with 10 μg/mL of dolutegravir, which resulted in a 5.03-fold increase in elastase release from unstimulated cells and a 2.55-fold increase in release of the protease in the case of FMLP/CB-stimulated cells.

### 2.3. Cytosolic Ca^2+^ Fluxes

These results are shown in Figure 3 as the median (IQR) magnitudes of alteration of cytosolic Ca^2+^ concentrations for neutrophils treated with dolutegravir in the presence or absence of extracellular Ca^2+^, as well as a single representative trace of 9–16 separate experiments. The addition of dolutegravir (10 μg/mL) resulted in an abrupt increase in neutrophil intracellular Ca^2+^ concentrations, which was abolished when the cells were pre-incubated in the absence of extracellular Ca^2+^. The median (IQR) cytosolic Ca^2+^ concentrations for the non-exposed neutrophils and the systems treated with dolutegravir in the presence or absence of extracellular Ca^2+^ were 431 (347:490), 753 (642:1069) and 200 (128:298) nM, respectively (9–16 replicates, *p* = 0.0015).

### 2.4. Cell Viability

Exposure of neutrophils to dolutegravir at 10 μg/mL for 15 min at 37 °C did not affect cellular viability. The mean percentage viability of the control cells and the cells treated with 10 μg/mL of dolutegravir were 98.3 (98:98.8) and 97.2 (97:97.5), respectively (*n* = 3 donors, with four replicates). Representative flow plots are shown in Figure 4.

### 2.5. Hemolytic Activity of Dolutegravir

These results are shown in Figure 5. Treatment of erythrocytes with dolutegravir caused hemolysis that was particularly evident at the higher concentrations (10 μg/mL and 20 μg/mL) of the drug when using the extended 2-h exposure time (*p* < 0.0000 for all concentrations). Prior addition of 5 mM of EGTA to the erythrocytes neutralized the hemolytic activity of dolutegravir (10 µg/mL), underscoring the dependence of this event on extracellular Ca^2+^.

## 3. Discussion

The findings of the current study have demonstrated a pro-inflammatory interaction between dolutegravir and neutrophils that manifests predominantly as a marked, spontaneous activation of the generation of ROS by the cells. This event is dependent on the presence of extracellular Ca^2+^ and intact NADPH oxidase, as revealed by the almost complete attenuation of dolutegravir-mediated activation of neutrophil ROS generation in the presence of the Ca^2+^-chelating agent, EGTA, or the selective NADPH oxidase inhibitor, DPI, in the cell-suspending media.

Similarly, exposure of neutrophils to dolutegravir also resulted in a dose-related, spontaneous release of the primary granule protease, elastase, which was considerably more prominent in the presence of the chemoattractant, FMLP. The stimulatory effects of dolutegravir on both neutrophil ROS generation and release of primary granules indicate that the differential Ca^2+^ thresholds for activation of these neutrophil activities [20] are exceeded in the presence of this agent.

Importantly, these pro-inflammatory interactions of dolutegravir were observed at non-cytotoxic concentrations (≤ 10 μg/mL) of the antiretroviral agent that are either attainable, or close to attainable, in the therapeutic setting. These have been reported to be in the range from 3.34–6.16 μg/mL following administration of tablets or suspensions at a daily dose of 50 mg [21].

With respect to the mechanisms underpinning the pro-inflammatory interactions of dolutegravir with neutrophils, the observed attenuation of pro-oxidative activity in the presence of EGTA implicates an influx of extracellular Ca^2+^ as being a key event, as mentioned above. This contention is strengthened by the observations that the addition of dolutegravir to neutrophils indeed resulted in an abrupt, marked, and sustained increase in cytosolic Ca^2+^ that was completely eliminated by inclusion of EGTA in the cell-suspending medium. This finding of dolutegravir-mediated Ca^2+^ influx into eukaryotic cells was confirmed using a simple hemolytic system as previously described by Bhuyan et al. [18]. Using this system, exposure to dolutegravir in a suspension of isolated human erythrocytes resulted in dose-related hemolysis that was also prevented by the inclusion of EGTA in the cell-suspending medium. The differential sensitivities of erythrocytes and neutrophils to dolutegravir-mediated cytotoxicity may reflect differences in the efficiencies of the intracellular Ca^2+^-handling machinery of the two cell types. In this context, we have recently reported that treatment of isolated human platelets results in an augmentation of their pro-adhesive/pro-inflammatory activities, also by a non-lytic mechanism that is dependent on the influx of extracellular Ca^2+^ [22]. In distinction to neutrophils, exposure of platelets to dolutegravir resulted in the sensitization of these cells to receptor-mediated stimulation, as opposed to direct activation. These observations underscore the differential effects of dolutegravir on various eukaryotic blood cell types, albeit by a common Ca^2+^-dependent mechanism.

With respect to the earlier study reported by Bhuyan et al., these authors described Ca^2+^-dependent oxidative stress, eryptosis, and hemolysis following an extended exposure of erythrocytes to dolutegravir at concentrations similar to those used here [18], while Wang et al. more recently described cytotoxic effects of a novel derivative of dolutegravir (5–20 μM) on human non-small cell lung cancer lines in vitro [19]. These effects of the novel derivative, but not the parent molecule, at an equivalent concentration of 10 μM, appeared to result from the elevated concentrations of cytosolic Ca^2+^ due to the inhibition of the endomembrane Ca^2+^-ATPase, oxidative stress, and induction of apoptosis [19]. From a mechanistic perspective, however, it is noteworthy that our study differs from those reported by Bhuyan et al. [18] and Wang et al. [19] in a number of significant respects, most notably differences in cell type, duration of exposure to dolutegravir (brief as opposed to prolonged), and lack of cytotoxicity in the case of neutrophils.

Given the physicochemical properties of dolutegravir, specifically the combination of lipophilicity and the ability to chelate divalent/trivalent metal ions [21,23,24], this agent is likely to possess ionophore-like properties. These, in turn, may enable dolutegravir to chelate extracellular Ca^2+^ cations and transport these across the plasma membrane to the cell cytosol where they activate/augment pro-inflammatory intracellular signaling mechanisms, such as PLC. Activation of PLC by dolutegravir, an event critically dependent on extracellular Ca^2+^, results in the assembly of the cytosolic and membrane-associated components of NADPH oxidase with consequent generation of ROS, including hydrogen peroxide that is converted to hypochlorous acid by MPO. The intracellular signaling pathway downstream of PLC includes IP3-mediated release of Ca^2+^ from intracellular storage vesicles and activation of PI3-kinase. The inhibitors of PLC, MPO, and PI-3-kinase cause marked attenuation of dolutegravir-mediated oxidant production. These findings suggest that dolutegravir-mediated activation of NADPH oxidase is triggered by the activation of PLC with associated mobilization of intracellular Ca^2+^ and that PI-3-kinase plays an important regulatory role during the activation of the oxidase.

It is well-established that weight gain and development of the metabolic syndrome are associated with an increased risk of many of the non-communicable diseases, such as cardiovascular disease, various cancers, and type 2 diabetes mellitus [25]. Weight gain in people living with HIV carries an even higher risk [26,27,28]. It has been reported that dolutegravir induces the release of IL-6 by adipocytes [29] which may potentiate the systemic inflammation associated with obesity. However, to what extent, if any, the pro-inflammatory interactions of dolutegravir with human neutrophils described in the current study may contribute to the pathophysiology of the unexplained, potentially harmful, weight gain in the setting of HIV infection is uncertain [6].

## 4. Materials and Methods

### 4.1. Ethical Statement 

The permission to undertake this study and to draw blood from healthy, adult human volunteers was granted by the Research Ethics Committee of the Faculty of Health Sciences, University of Pretoria, and the study was performed in full compliance with the World Medical Association Declaration of Helsinki 2013 (Approval Nos. 116/2017 and 605/2018). Prior written informed consent was obtained from all blood donors, each of whom underwent a routine health check (including measurement of blood pressure) by an experienced, qualified nurse prior to the blood draw.

### 4.2. Chemicals and Reagents

Dolutegravir, (4R,12aS)-*N*-[(2,4-difluorophenyl)methyl]-3,4,6,8,12,12a-hexahydro-7-hydroxy-4-methyl-6,8-dioxo-2*H*-pyrido [1′,2′:4,5]pyrazino [2,1-b][1,3]oxazine-9-carboxamide, was purchased from the Cayman Chemical Company (Ann Arbor, MI, USA) and dissolved in dimethyl sulfoxide (DMSO, 0.05–0.1%) at final concentrations of 2.5–10 µg/mL (approximately 5–20 µM).

Unless indicated, all other chemicals and reagents were purchased from Sigma-Aldrich (St Louis, MO, USA).

### 4.3. Preparation of Neutrophils

Neutrophils were isolated from heparinized venous blood (5 units of preservative-free heparin per ml of blood) from 22 healthy volunteers (mean age ± standard deviation, 36 ± 11). Neutrophils were separated from mononuclear leukocytes by centrifugation on Histopaque-1077 (Sigma-Aldrich) cushions at 400× *g* for 25 min at room temperature. The resultant pellets were suspended in a phosphate-buffered saline (PBS, 0.15 M, pH 7.4) and sedimented with 3% gelatine to remove most of the erythrocytes. Following centrifugation (280× *g* at 4 °C for 10 min), residual erythrocytes were removed by selective lysis with 0.83% ammonium chloride at 4 °C for 10 min. The neutrophils, which were routinely of high purity (>90%) and viability, as determined by flow cytometric procedures, were resuspended to 1 × 10^7^ /mL in a phosphate-buffered saline (PBS, pH 7.4) and held on ice until used.

### 4.4. Measurement of Reactive Oxygen Species

These were measured using lucigenin (bis-*N*-methylacridinium nitrate)-enhanced and luminol (5-amino-2,3-dihydro-1,4-phthalazine dione)-enhanced chemiluminescence (CL) procedures that predominantly detect superoxide and reactive oxygen species (ROS) generated by the myeloperoxidase/H_2_O_2_/halide system, respectively [30]. Briefly, neutrophils (2 × 10^5^ /mL) were pre-incubated for 10 min at 37 °C in 900 mL of Hanks’ balanced salt solution (HBSS) containing either lucigenin (0.2 mM) or luminol (0.1 mM), followed by the addition of either 1 μL of DMSO (resting control) or dolutegravir (2.5–10 μg/mL) alone, and the CL responses were recorded using a Lumac Biocounter (Model 2010, Lumac Systems Inc, Titusville, FL, USA). The final volume in each vial was 1 mL, and the results, which are expressed in relative light units (RLU), are the peak values for treated systems that reached 40–60 s after the addition of the drug.

In an additional series of experiments, the effects of the following reagents on dolutegravir-mediated activation of luminol-enhanced CL were investigated: (i) the extracellular Ca^2+^-chelating agent, EGTA [ethylene glycol-bis(β-aminoethyl ether)-*N,N,N’,N’*-tetraacetic acid tetrasodium salt, 5 mM]; (ii) an inhibitor of the neutrophil superoxide-generating NADPH oxidase, diphenyleneiodonium chloride (DPI, 5 µM, dissolved in DMSO); (iii) an inhibitor of phospholipase C (PLC), U73122 (5 μM, dissolved in DMSO); (iv) an inhibitor of the neutrophil myeloperoxidase (MPO), sodium azide (50 μg/mL); and (vi) an inhibitor of phosphatidylinositol-3-kinase (PI3 Kinase), wortmannin (1 μM, dissolved in DMSO). EGTA was added to the neutrophil suspensions 1 min prior to dolutegravir, while the inhibitors or the DMSO solvent control were present throughout the 10 min pre-incubation period.

### 4.5. Elastase Release

Neutrophil degranulation was measured according to the extent of release of the primary granule enzyme, elastase. Neutrophils were incubated at a concentration of 5 × 10^5^/mL in HBSS with and without dolutegravir (2.5–10 μg/mL) or 1 μL of DMSO (resting control) for 10 min at 37 °C. The chemoattractant, N-formyl-L-methionyl-L-leucyl-L-phenylalanine (FMLP, 0.1 μM), in combination with cytochalasin B (1 μM) was then added to the cells and incubated for a further 10 min at 37 °C. The tubes were then transferred to an ice bath, followed by centrifugation at 400× *g* for 5 min to pellet the cells. The neutrophil-free supernatants were then assayed for elastase using a micromodification of a standard colorimetric procedure [31]. Briefly, 125 μL of supernatant was added to the elastase substrate, *N*-succinyl-*L*-alanyl-*L*-alanyl-*L*-alanine-*p*-nitroanilide (3 mM in DMSO), in 0.05 M Tris-HCl (pH 8), and elastase activity was monitored spectrophotometrically at a wavelength of 405 nm.

### 4.6. Cytosolic Ca^2+^ Fluxes

These were measured spectrofluorimetrically using Fluo-8 acetoxymethyl ester (Fluo-8-AM) as the Ca^2+^-sensitive indicator of cytoplasmic Ca^2+^ [32]. Neutrophils (1 × 10^7^ /mL) suspended in PBS were pre-loaded with Fluo-8/AM (2 µM) for 25 min at 37 °C, washed and resuspended in PBS, and held on ice until used. For the measurement of intracellular Ca^2+^ fluxes, the neutrophils were transferred to indicator-free HBSS (pH 7.4 containing 1.25 mM Ca^2+^). The Fluo-8/AM-loaded cells were then pre-incubated for 8 min in a 37 °C water bath, followed by transfer to a disposable reaction cuvette, which was maintained at 37 °C in a Hitachi 650–10 S fluorescence spectrophotometer (Hitachi Ltd., Tokyo, Japan) with excitation and emission wavelengths set at 485 nm and 525 nm, respectively. Importantly, at these wavelengths, no evidence of dolutegravir-induced autofluorescence was detected. The final volume in each cuvette was 3 mL containing 3 × 10^6^ neutrophils. After a stable wavelength was obtained (1 min), dolutegravir (10 µg/mL) was added to the cuvettes and alterations in the fluorescence intensity were monitored over a 5 min period. Cytoplasmic Ca^2+^ concentrations were calculated as described previously [32].

In an additional series of experiments, neutrophils were suspended in Ca^2+^-free HBSS plus EGTA (3 mM), an extracellular Ca^2+^-chelating agent, to investigate dolutegravir-mediated alterations in neutrophil cytoplasmic Ca^2+^ concentrations in the absence of extracellular Ca^2+^.

### 4.7. Cell Viability

This was measured after a 15 min incubation period following the addition of DMSO (resting control) or dolutegravir (10 μg/mL) to neutrophils (1 × 10^6^ /mL) using a flow cytometric propidium iodide-based dye exclusion assay. The cells were incubated for 10 min with propidium iodide (50 μg/mL, DNA Prep-Stain, Beckman Coulter, Miami, FL, USA), and cell viability was assessed using a FC500 flow cytometer (Beckman Coulter, Miami, FL, USA). The results are expressed as the percentages of viable cells. The analysis was performed using FlowJo version 10.8.1 for Windows (BD BioSciences, Franklin Lakes, NJ, USA). Bisector gates were applied to distinguish between propidium iodide negative (viable cells) and propidium iodide positive populations (apoptotic cells) in the control. The same gates were applied to all subsequent samples.

### 4.8. Hemolytic Assay

To further explore the potential of dolutegravir to mobilize extracellular Ca^2+^, the effects of this agent on erythrocyte membrane integrity in the absence and presence of the divalent cation were determined using a hemolytic assay. Erythrocytes were isolated from heparinized venous blood (5 units of preservative-free heparin per ml of blood) from the healthy volunteers. Erythrocytes were separated from leukocytes by centrifugation on Histopaque-1077 (Sigma-Aldrich) cushions at 400× *g* for 25 min (min) at room temperature. Two milliliters of erythrocytes were washed 3 times and re-suspended to 0.5% in PBS. The cells (final concentration 0.05%) were then co-incubated with DMSO (resting control) or dolutegravir (5–20 µg/mL) for either 20 min or 2 h at 37 °C. Intact erythrocytes were removed by centrifugation and the supernatants were assayed spectrophotometrically at 405 nm for hemoglobin content. The results are expressed as % of hemolysis. To probe the involvement of extracellular Ca^2+^, the effects of EGTA (5 mM) on dolutegravir (10 µg/mL)-mediated hemolytic activity were also investigated.

### 4.9. Statistical Analysis

Except for the results of the Fluo-8 fluorescence experiments, some of which are presented as representative traces, the results of each series of experiments are presented as the median values and inter quartile ranges (IQRs). The number of different donors (n) used in each series of experiments and the number of replicates for each experiment are shown in the text and figure legends. Levels of statistical significance are determined by comparing the absolute values for each drug-treated system with the corresponding values for the relevant drug-free control systems for each assay, using the Wilcoxon sign rank test. Box and Whisker plots depict the 75th percentile (upper hinge of box), median (line in box) and 25th percentile (lower hinge of box). Whiskers indicate the upper (maximum) and lower (minimum) adjacent values of the dataset. The analysis was performed using Stata 17.0 SE (StataCorp, College Station, TX, USA) and a *p* value of < 0.05 was considered statistically significant.

## 5. Conclusions

Given the widespread use of dolutegravir, a better understanding of the potential clinical impact of its pro-inflammatory effects on neutrophils, possibly related to its ionophore-like properties, merits further investigation.

## Figures and Tables

**Figure 1 molecules-27-09057-f001:**
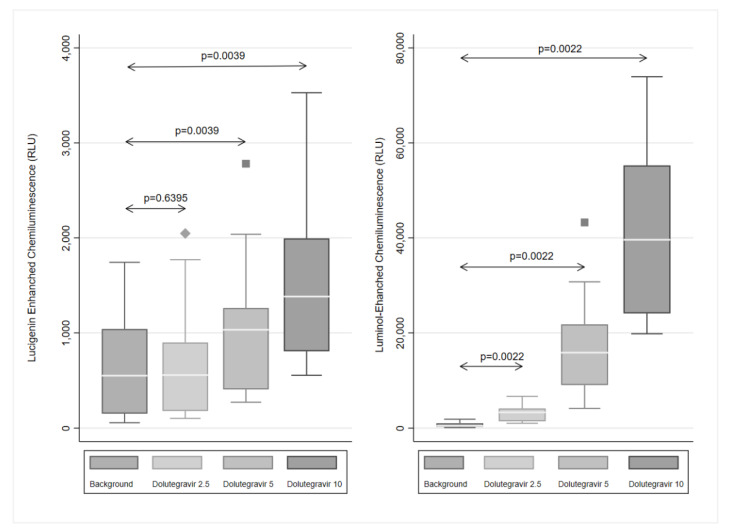
Effects of dolutegravir (2.5–10 μg/mL) on the spontaneous lucigenin- and luminol-enhanced chemiluminescence responses of neutrophils. The results are expressed as the median peak chemiluminescence values in relative light units (RLU) measured 30–50 s after the addition of dolutegravir using Box and Whisker plots. Lucigenin-enhanced chemiluminescence (left graph): *n* = 5 donors with 9 replicates for each drug concentration and control system, and luminol-enhanced chemiluminescence (right graph): *n* = 6 with 30 replicates. “Background” refers to the unstimulated DMSO-treated control. *p* = 0.6395–*p* = 0.002 for comparison of the dolutegravir-treated systems with the unstimulated control system.

**Figure 2 molecules-27-09057-f002:**
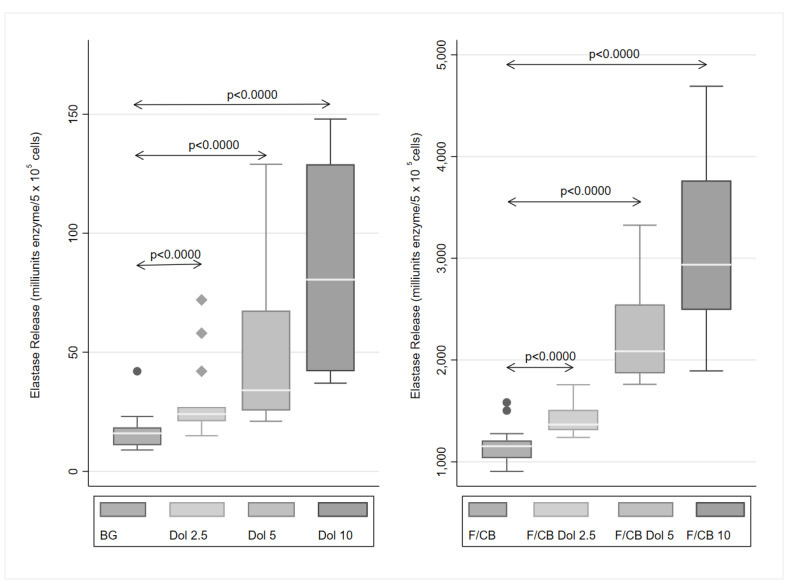
Effects of dolutegravir (Dol, 2.5–10 μg/mL) on the release of elastase from unstimulated neutrophils (spontaneous release), as well as from cells activated with FMLP (0.1 μM)/cytochalasin B (1 μM) (F/CB). The results (*n* = 4 donors with 5 replicates for each experiment) are expressed as Box and Whisker plots (milliunits/5 × 10^5^ cells). “BG” (Background) refers to the unstimulated DMSO-treated control. *p* < 0.0000 for comparison of the dolutegravir-treated systems with the relevant control system.

**Figure 3 molecules-27-09057-f003:**
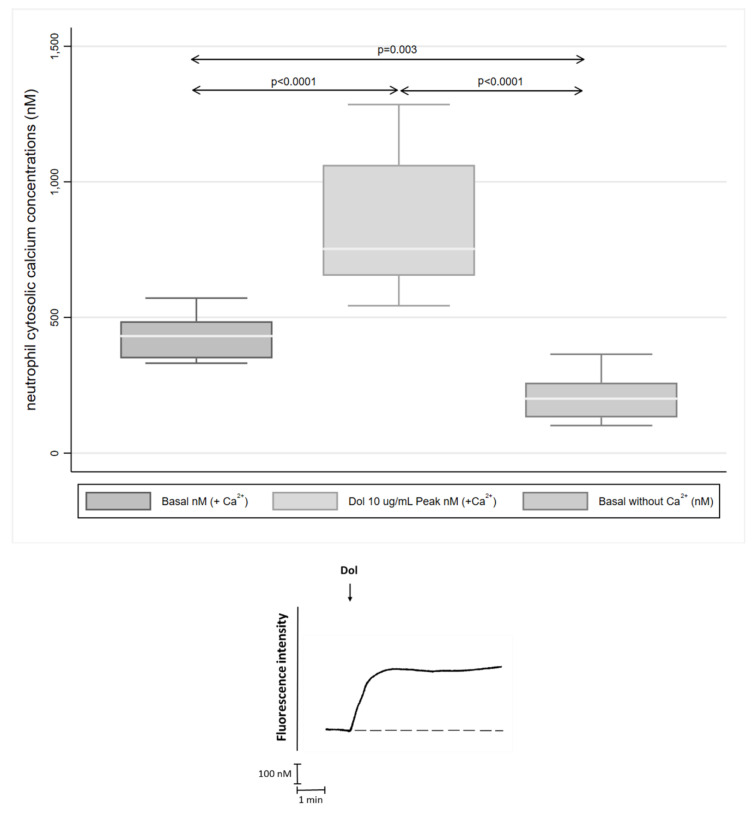
Effects of dolutegravir on neutrophil cytosolic Ca^2+^concentrations. Dolutegravir (Dol, 10 μg/mL) as denoted by the arrow (↓) was added to neutrophils suspended in Ca^2+^ -replete (-) and –depleted (- - - -) medium. An abrupt and sustained increase in cytosolic Ca^2+^ was observed following the addition of dolutegravir to the neutrophils suspended in Ca^2+^-replete HBSS, which was abolished in a Ca^2+^-free medium. The results of the entire series of experiments (16 separate experiments; *n* = 7 individual donors) are shown in the upper panel as the basal and peak cytosolic Ca^2+^ concentrations following the addition of dolutegravir to the neutrophils suspended in a Ca^2+^-replete medium and the basal Ca^2+^ concentration in the absence of extracellular Ca^2+^. Representative traces from a single experiment are shown in the lower panel. The tracing in the lower panel depicts a single experiment that is representative of those in the series of 9–16 replicates.

**Figure 4 molecules-27-09057-f004:**
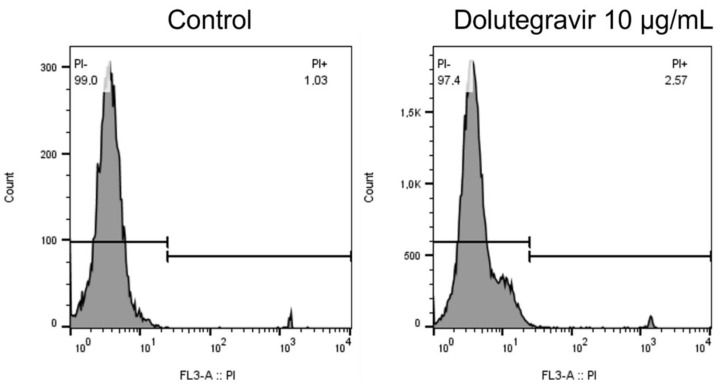
Cell viability is comparable between the control and dolutegravir-treated cells.

**Figure 5 molecules-27-09057-f005:**
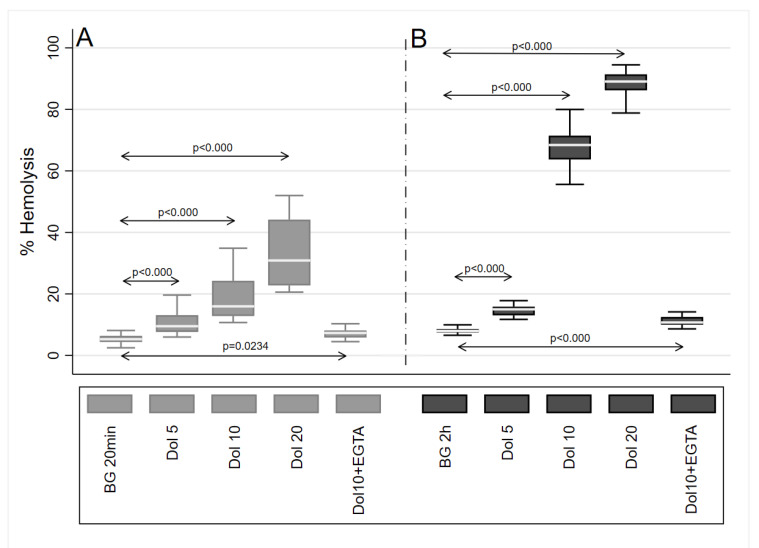
The hemolytic effects of dolutegravir (5–20 μg/mL) at exposure times of 20 min (**A**) and 2 h (**B**), respectively. The results (*n* = 3 donors with 5 replicates) are expressed as median percentage hemolysis in Box and Whisker plots.

## Data Availability

Data can be made available upon request.

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
