# Peer review of "Pro-Inflammatory Interactions of Dolutegravir with Human Neutrophils in an In Vitro Study"

_molecules, 2022, doi:10.3390/molecules27249057_

Round 1
Reviewer 1 Report
This paper shows a dose-dependent, dolutegravir-induced activation of primary neutrophils that is dependent on extracellular calcium signaling. The authors suggest this may provide a link to the documented side-effects of weight-gain and metabolic syndrome in HIV patients taking dolutegravir.
The paper is straight-forward and well written and presented. I have only minor issues to address:
Since this study seeks to address a possible link between dolutegravir, inflammation and weight-gain, it may benefit from noting that doultegravir increased IL-6 release in apidpocytes in vitro (https://doi.org/10.1016/j.lfs.2022.120948). If other studies have noted similar pro-inflammatory cytokine expression induced by dolutegravir, a brief discussion may help bolster the paper’s thesis.
While the extent of the boxes in the box and whisker plots is defined in methods, it seems the extent of whiskers is not defined anywhere. Presumably they extend to 95% confidence intervals and not maximum and minimum values since outliers are present in some figures, but a reader shouldn’t have to guess. Please define the extent of the whiskers. Additionally, in some figure legends the n is defined by number of donors or number of experiments while it is not defined in others (e.g. Figure 1). Please state what you are using to define the n in each case.
Figure 3 – the main body text states the trace represents 4 single experiments while the figure legend says 1. Please clarify.
Author Response
Reviewer 1:
- We thank the expert Reviewer for bringing to our attention that dolutegravir has been shown to increase IL-6 release by adipocytes. In recognition of this, we have included the following sentence and reference in the Discussion on page 10, lines 365-367 “It has been reported that dolutegravir induces the release of IL-6 by adipocytes [32] which may potentiate the systemic inflammation associated with obesity.”
- The extent of the whiskers in the “Box and Whiskers” plots indicates the upper (maximum) and lower (minimum) adjacent values of the dataset and this has been defined in the Materials and Methods section on page 4, lines 191-193.
- The number ‘n’ for Figures 1 and 2 has been defined in the Figure legends on page 5, line 210 and page 6, line 243 respectively.
- The legend to figure 3 has been clarified to indicate that the tracing depicts a single experiment that is representative of those in the series with 9-16 replicates (Page 8, lines 266-267.
In conclusion, we are most grateful for the points raised by the expert Reviewers and trust that the revised version of the manuscript will be acceptable for publication in Molecules.
Yours sincerely
Prof. AJ Theron
Reviewer 2 Report
This is an interesting paper on a relevant topic. However, critical controls are missing from all experiments, making it hard to interpret the data.
- The compound studied needs to be compared with cells treated with a vehicle control. In this case the impact of DMSO. DMSO, even at low concentrations, can have an effect on neutrophil function. Therefore, experiments need to be compared to DMSO only.
- Please show representative tracing/flow cytometry plots.
- It is not clear why hemolysis was assessed in this manuscript which mainly focusses on neutrophils.
- It is also odd how neutrophil viability was assessed only after 15min, while hemolysis was assessed after 2h. Please repeat viability experiments for longer timepoints.
- the methods don't have details on how erythrocytes were isolated
Author Response
Reviewer 2:
- The control experiments for each series were conducted using DMSO-treated neutrophils. This has been emphasized in the revised manuscript in the Materials and Methods section on page 3, lines 110 and 129 and page 4, line 159 and 177, as well as in the legends to Figures 1 and 2 on page 5, line 212 and page 6, line 244, respectively.
- The flow cytometry plots representing experiments to evaluate the effects of dolutegravir on neutrophil viability have been included in the manuscript as Figure 4, page 8.
- The assays to determine the effects of dolutegravir on erythrocytes were included in the current study in order to evaluate the responses of various eukaryotic blood cell types to dolutegravir. This has been described in the Discussion on page 10, lines 322-334.
- The assays to determine the effects of dolutegravir on neutrophil viability were assessed after 15 minutes as the incubation times of neutrophils with dolutegravir for the different experiments performed in the current study varied from 5 to 10 min.
- The methods used to isolate human erythrocytes have now been described in greater detail in the Materials and Methods section on page 4, lines 170-182.
In conclusion, we are most grateful for the points raised by the expert Reviewers and trust that the revised version of the manuscript will be acceptable for publication in Molecules.
Yours sincerely
Prof. AJ Theron
Round 2
Reviewer 2 Report
Thank you for responding and addressing the concerns.